# Insight into the Systematics of Novel Entomopathogenic Fungi Associated with Armored Scale Insect, *Kuwanaspis howardi* (Hemiptera: Diaspididae) in China

**DOI:** 10.3390/jof7080628

**Published:** 2021-08-02

**Authors:** Xiu-Lan Xu, Qian Zeng, Yi-Cong Lv, Rajesh Jeewon, Sajeewa S. N. Maharachchikumbura, Dhanushka N. Wanasinghe, Kevin D. Hyde, Qian-Gang Xiao, Ying-Gao Liu, Chun-Lin Yang

**Affiliations:** 1National Forestry and Grassland Administration Key Laboratory of Forest Resources Conservation and Ecological Safety on the Upper Reaches of the Yangtze River, Sichuan Agricultural University, Chengdu 611130, China; xuxiulanxxl@126.com (X.-L.X.); zq1573037145@163.com (Q.Z.); Lvyicong0616@126.com (Y.-C.L.); lyg092764@163.com (Y.-G.L.); 2Sichuan Province Key Laboratory of Ecological Forestry Engineering on the Upper Reaches of the Yangtze River, Sichuan Agricultural University, Chengdu 611130, China; 3Research Institute of Forestry, Chengdu Academy of Agricultural and Forestry Sciences, Chengdu 611130, China; xiaoqg1992@163.com; 4Department of Health Sciences, Faculty of Medicine and Health Sciences, University of Mauritius, Reduit 80837, Mauritius; r.jeewon@uom.ac.mu; 5School of Life Science and Technology, University of Electronic Science and Technology of China, Chengdu 611731, China; sajeewa83@yahoo.com; 6CAS Key Laboratory for Plant Diversity and Biogeography of East Asia, Kunming Institute of Botany, Chinese Academy of Sciences, Kunming 649201, China; dnadeeshan@gmail.com; 7Center of Excellence in Fungal Research, Mae Fah Luang University, Chiang Rai 57100, Thailand; kdhyde3@gmail.com

**Keywords:** 2 new taxa, bamboo, entomopathogens, Nectriaceae, Podonectriaceae, scale insect

## Abstract

This study led to the discovery of three entomopathogenic fungi associated with *Kuwanaspis howardi*, a scale insect on *Phyllostachys heteroclada* (fishscale bamboo) and *Pleioblastus amarus* (bitter bamboo) in China. Two of these species belong to *Podonectria*: *P. kuwanaspidis* X.L. Xu & C.L. Yang sp. nov. and *P*. *novae-zelandiae* Dingley. The new species *P. kuwanaspidis* has wider and thicker setae, longer and wider asci, longer ascospores, and more septa as compared with similar *Podonectria* species. The morphs of extant species *P. novae-zelandiae* is confirmed based on sexual and asexual morphologies. Maximum likelihood and Bayesian inference analyses of ITS, LSU, SSU, *tef1-α*, and *rpb2* sequence data provide further evidence for the validity of the two species and their placement in Podonectriaceae (Pleosporales). The second new species, *Microcera kuwanaspidis* X.L. Xu & C.L. Yang sp. nov., is established based on DNA sequence data from ITS, LSU, SSU, *tef1-α*, *rpb1*, *rpb2*, *acl1*, *act*, *cmdA*, and *his3* gene regions, and it is characterized by morphological differences in septum numbers and single conidial mass.

## 1. Introduction

*Podonectria* was introduced by Petch [1] to accommodate species of *Ophionectria*, which are parasitic on scale insects and have thick-walled asci, long, multiseptate ascospores, and a tetracrium-like conidial stage. Ten species are listed in Index Fungorum [2]. The type species, *Podonectria coccicola* (Ellis and Everh.) Petch was transferred from *Ophionectria coccicola* (Ellis & Everh.) Berl. & Voglino and is associated with the scale insects *Aonidiella aurantia* (Maskell), *Aspidiotus perniciosus* (Comstock), *Chrysomphalus aonidum* (Linnaeus), *Lepidosaphes beckii* (Newman), *L. gloverii* (Packard), *Leucapsis* sp., *Parlatoria pergandii* Comstock, *P. ziziphi* Lucas, and *Unaspis citri* (Comstock) which are mainly found on Rutaceae [1,3,4,5]. *Puttemansia aurantii* (Henn.) Höhn, which was initially found from the type specimen of the asexual morph *Tetracrium aurantii* Henn. associated with scale insect *Parlatoria ziziphi* on *Citrus aurantium* L., was also transferred to *Podonectria* as *P. aurantii* (Henn.) Petch [1]. A new species collected from *Lepidosaphes* sp. on *Citrus nobilis* Lour. was named as *Podonectria echinata* [1]. Additionally, two new species, *P. gahnia* Dingley and *P. novae-zelandiae* Dingley, were reported by Dingley (1954) from scale insects in New Zealand [3], followed by a new fungus *P. tenuispora* Dennis collected from *Lepidosaphes ulmi* (Linnaeus) on *Calluna vulgaris* (L.) Hull [6]. Subsequently, Rossman transferred *Ophionectria coccorum* Petch, associated with *Fiorinia juniperi* Kuwana, and *Lasiosphaeria larvaespora* Cooke & Massee on an undetermined scale insect to *Podonectria*, viz. *P. coccorum* (Petch) Rossman and *P. larvaespora* (Cooke & Massee) Rossman [7]. The species *Trichonectria bambusicola* Rehm was referred as *P. bambusicola* (Rehm) Piroz. on account of scolecosporous ascospores and tetracrium-like conidia by Pirozynski [8]. However, *Podonectria bambusicola* was excluded because of its occurrence on living leaves of bamboo rather than scale insects and remained an unclassified loculoascomycete [4]. Rossman published a monograph on *Podonectria* and accepted eight species [4]. An examination of the type specimen of *T. bambusicola* further revealed that this was a synonym of *Uredinophila erinaceae* (Rehm) Rossman [9]. The genus *Podonectria* was characterized by fleshy, white to brown, uninoculated ascomata with bitunicate asci and long, multiseptated ascospores associated with scale insects [4]. Spatafora et al. [10] transferred the previously reported species *Podonectria cicadellidicola* Kobayasi & Shimizu and *P. citrina* Kobayasi & Shimizu to *Ophiocordyceps* supported by the previous phylogenetic analyses presented in Quandt et al. [11]. Yang et al. [12] found *P. sichuanensis* C.L. Yang & X.L. Xu parasitic around the ascomata of *Neostagonosporella sichuanensis* C.L. Yang, X.L. Xu & K.D. Hyde on *Phyllostachys heteroclada* Oliv.

*Microcera* (Nectriaceae, Hypocreales), typified by *Microcera coccophila* Desm. and known as the “red-headed fungus”, is mostly parasitic on scale insects with fusarium-like asexual morphs. The genus has been considered as a synonym in major taxonomic revisions of *Fusarium* Link [13,14,15,16,17]. A multilocus phylogenetic approach was subsequently applied to identify species in the fusarium-like clade since morphological identification was difficult [18,19,20]. Gräfenhan et al. [18] resurrected *Microcera* based on DNA sequence data and accepted four *Microcera* species, viz. *M. coccophila*, *M. diploa* (Berk. & M.A. Curtis) Gräfenhan & Seifert, *M. rubra* Gräfenhan & Seifert, *M. larvarum* (Fuckel) Gräfenhan, Seifert & Schroers. Lombard et al. [19] further investigated phylogenetic relationships of *Microcera* based on DNA sequence data and reported that it constitutes a lineage distantly related to *Fusarium* but closely related to *Fusicolla* and *Macroconia*.

Armored scale insects (Hemiptera: Coccomorpha: Diaspididae) are major economic pests on agriculture and forestry plants, especially on fruit trees and vegetables. Diaspididae is the largest family of scale insects with 421 accepted genera and four subfamilies recognized, viz. Ancepaspidinae Borchsenius, Furcaspidinae Balachowsky, Diaspidinae Targioni Tozzetti, and Aspidiotinae Westwood. by Normark et al. [21]. The grass-feeding species, *Kuwanaspis* MacGillivray, which is classified into subtribe Fioriniina Targioni Tozzetti under tribe Diaspidini Targioni Tozzetti within subfamilies Diaspidinae, are harmful to bamboo [22,23]. During our investigations of microfungi associated with bamboo in Sichuan Province, two *Podonectria* species and a *Microcera* species were isolated in association with the armored scale insect *Kuwanaspis howardi* (Cooley) on native bamboo plants *Phyllostachys heteroclada* and *Pleioblastus amarus* (Keng) Keng. Morphological characteristics coupled with phylogenetic analyses of the combined ITS, LSU, SSU, *tef1-α*, and *rpb2* sequence dataset support the validity of the *P. kuwanaspidis* X.L. Xu & C.L. Yang sp. nov. and *P. novae-zelandiae* Dingley and their placement in Podonectriaceae, Pleosporales. The fusarium-like species *Microcera kuwanaspidis* is distinguished from similar species based on the sequences’ differences, mainly in the *tef1-α*, *acl1*, *act*, *cmdA*, *rpb1*, and *his3* regions. This is the first record of these taxa associated with scale insects in China. The taxa are compared with allied species, and comprehensive descriptions and micrographs are provided.

## 2. Materials and Methods

### 2.1. Specimen Collection and Morphological Study

During spring to autumn from 2018 to 2021, the specimens were collected from the bamboo forests located in Ya’an City and a neighboring county (Sichuan Province, China), where the environment is characterized by river valley terraces and intermountain basins and a subtropical monsoon humid climate with abundant natural resources, and it is the transition zone from Qinghai-Tibet Plateau to Chengdu Plain. Specimens documented with host, locality, time, and distribution of taxa were returned to the laboratory in suitable containers separately with the collection detail tag, and the substrate with fruiting bodies was checked following the methods described in Senanayake et al. [24]. The fungi were isolated into pure culture using single conidium obtained from sporodochia and single ascospore from ascomata parasitic on *Kuwanaspis howardi* following the isolation via spore suspension detailed in Chomnunti et al. [25]. The spore suspension was sucked into a Pasteur pipette, small drops were placed on isolation media (potato dextrose agar, PDA) in an incubator (20 °C). Then the plates were examined for single germinated spores under a dissecting microscope, and germinating spores were transferred separately to at least three new PDA plates. After incubation on PDA plates at 20 °C for 20 to 40 days depending on the growth rate, colonies were examined for their diameter, shape, and appearance. Ascomata and sporodochia were observed and photographed using a dissecting microscope NVT-GG (Shanghai Advanced Photoelectric Technology Co. Ltd., Shanghai, China) fitted with a VS-800C micro-digital camera (Shenzhen Weishen Times Technology Co. Ltd., Shenzhen, China). Dimensions of asci, ascospores, pseudoparaphyses, hairs, ascomata wall, conidia, conidiophores, and numbers of septa were based on field samples and were photographed using an Olympus BX43 compound microscope fitted with an Olympus DP22 digital camera in association with ACDSee v3.1 software. Measurements were made using Tarosoft^®^ Image Frame Work v.0.9.7 (Tarosoft (R), Nontha Buri, Thailand). Lactophenol cotton blue reagent was used to observe the number of septa. The gelatinous appendage was observed in Black Indian ink. The type specimens were deposited at the Herbarium of Sichuan Agricultural University, Chengdu, China (SICAU). The ex-type cultures were deposited at the Culture Collection in Sichuan Agricultural University (SICAUCC), and MycoBank numbers are registered (http://www.MycoBank.org, accessed on 10 January 2021).

### 2.2. DNA Extraction, Amplification and Sequencing

Total genomic DNA was extracted from mycelia grown on PDA at 20 °C for 30 days, using the Plant Genomic DNA extraction kit (Tiangen, China). The internal transcribed spacer (ITS), the partial large subunit nuclear rDNA (LSU), the partial small subunit nuclear rDNA (SSU), translation elongation factor 1-alpha (*tef1-**α*), the RNA polymerase II second largest subunit (*rpb2*), the large subunit of the ATP citrate lyase (*acl1*), the RNA polymerase II largest subunit (*rpb1*), β-tubulin (*tub2*), histone H3 (*his3*), translation elongation factor 1-alpha (*tef1-α*), calmodulin (*cmdA*), and actin (*act*) regions were amplified with primer pairs ITS5/ITS4 [26], LR0R/LR5 [27], NS1/NS4 [26], EF1-983F/EF1-2218R [28], fRPB2-5F/fRPB2-7cR [29], acl1-230up/acl1-1220low, RPB1-Ac/RPB1-Cr, T1/CYLTUB1R, CYLH3F/CYLH3R, EF1-728F/EF2, CAL-228F/CAL2Rd, and ACT-512F/ACT-1Rd [19], respectively.

Polymerase chain reaction (PCR) was performed in 25 μL reaction mixture containing 22 μL Master Mix (Beijing TsingKe Biotech Co., Ltd., Beijing, China), 1 μL DNA template, 1 μL each primer (10 μM). The amplification reactions were performed as described by Gräfenhan et al. [18], Lombard et al. [19], Dai et al. [30], and Wanasinghe et al. [31]. PCR products were sequenced at TsingKe Biological Technology Co., Ltd., Chengdu, China. The newly generated sequences were deposited in GenBank.

### 2.3. Phylogenetic Analyses

To infer relationships of our *Podonectria* taxa, a combined ITS, LSU, SSU, *tef1-α*, and *rpb2* sequences dataset was used to construct the phylogenetic tree. For *Microcera* taxa, a combined ITS, LSU, *tef1-α*, *rpb1*, *rpb2*, *acl1*, *act*, *tub2*, *cmdA*, and *his3* sequences dataset was used. Taxa used for phylogenetic analyses were selected based on BLAST searches and recent publications (Table 1 and Table 2). DNA alignments were performed using MAFFT v.7.429 online service [32], and ambiguous regions were excluded using BioEdit version 7.0.5.3 [33]. Phylogenetic trees were inferred with maximum likelihood (ML) and Bayesian inference (BI), according to the details described in Xu et al. [34]. The finalized alignments and trees were deposited in TreeBASE (http://www.treebase.org, accessed on 10 January 2021), submission ID: 27547 and 27549, respectively.

## 3. Results

### 3.1. Phylogenetic Analyses

Phylogenetic analyses of a combined five-gene dataset (ITS, LSU, SSU, *tef1-α*, *rpb2*) comprised 62 taxa, and the tree is rooted with *Tubeufia javanica* Penz. & Sacc. (MFLUCC 12-0545) and *T. chiangmaiensis* Boonmee & K.D. Hyde (MFLUCC 11-0514) (Tubeufiaceae, Tubeufiales). The alignment contained 5721 characters (LSU = 1046, ITS = 821, SSU = 1176, *tef1-α* = 1507, *rpb2* = 1171), including gaps. The best scoring RAxML tree with a final likelihood value of −40,064.587233 is presented. The matrix had 2539 distinct alignment patterns, with 46.29% of undetermined characters or gaps. Estimated base frequencies were as follows: A = 0.244598, C = 0.248213, G = 0.265992, T = 0.241197, with substitution rates AC = 1.565487, AG = 3.743698, AT = 1.774643, CG = 1.114196, CT = 7.131582, GT = 1.000000. The gamma distribution shape parameter α = 0.240760, and the Tree-Length = 3.685780.

Phylogenetic trees generated from ML and BI analyses were similar in overall topologies. Phylogeny from the combined sequence data analysis indicates that all the Pleosporalean families are monophyletic with strong bootstrap support values (Figure 1). Three species grouped with taxa in *Podonectria* with 100% ML and 1.00 BYPP support. A species (SICAUCC 21-0004, SICAUCC 21-0005) clustered with *P. novae-zelandiae* in a clade with 99% ML and 1.00 BYPP statistical support. Our novel species *P. kuwanaspidis* constitutes a moderately supported independent lineage (82% ML/-- BYPP statistical support) between *P. novae-zelandiae* and *P. coccicola*.

DNA sequences of four known species of *Microcera* and our new taxon, *M. kuwanaspidis*, were used in the analyses. The combined dataset comprised 24 taxa within Nectriaceae and two outgroup taxa in Tilachlidiaceae (Table 2). The alignment contained 7447 characters (ITS = 638, LSU = 831, *acl1* = 1041, *act* = 673, *cmdA* = 778, *his3* = 530, *rpb1* = 741, *rpb2* = 874, *tef1-α* = 631, *tub2* = 710), including gaps. The tree is rooted with *Tilachlidium brachiatum* (Batsch) Petch (CBS 363.97, CBS 505.67). The best scoring RAxML tree with a final likelihood value of −50,074.064664 is presented. The matrix had 3327 distinct alignment patterns, with 28.65% of undetermined characters or gaps. Estimated base frequencies were as follows: A = 0.233349, C = 0.272026, G = 0.255053, T = 0.239571, with substitution rates AC = 1.285610, AG = 3.696293, AT = 1.292257, CG = 0.967004, CT = 6.115076, GT = 1.000000. The gamma distribution shape parameter α = 0.261457, and the Tree-Length = 2.372705. In the concatenated phylogenetic analyses of ML and BI, all species of *Microcera* analyzed clustered in a well-supported clade (ML = 100%, BYPP = 1.00) with a close affinity to *Fusicolla* and *Macroconia* (Figure 2). *Microcera kuwanaspidis* is related to *M. coccophila* in a subclade with 100% ML and 1.00 BYPP statistical support.

### 3.2. Taxonomy

Podonectriaceae H.T. Dao & Rossman, Mycological Progress 15(5): 47 (2016) amended.

MycoBank number: MB 815827

Type genus: *Podonectria* Petch, Trans. Br. mycol. Soc. 7(3): 146 (1921).

*Parasitic* fungus on scale insects, other fungi, or substrates previously colonized by other fungi. *Sexual morph*: *Stromata* byssoid, well-developed or scant, white to brown or dark-brown. *Ascomata* solitary or aggregated, superficial on or immersed in the stroma, globose to subglobose, obpyriform or ovoid, cream white to light yellow, or brown to dark brown, covered with hairs or absent. The *hamathecium* comprises numerous reticulate, filiform, septate, branched, pseudoparaphyses. *Asci* 8-spored, bitunicate, long clavate to cylindric. *Ascospores* long clavate to long cylindric, or vermiform, multiseptate. *Asexual morph*: Tetracrium-like. *Sporodochia* formed directly on cushion-shaped, white, orange, or brown, and hard stroma. *Conidiophores* moniliform. *Conidia* usually 1–4 “arms”, narrowed toward the apex, joined at the basal cell, multiseptate.

Notes: The family Podonectriaceae was introduced to accommodate *Podonectria* by Dao et al. [5], in which descriptions of conidia, ascomata, asci, and ascospores were lacking. Here we emend those descriptions and the habitats of Podonectriaceae with the inclusion of fungi or substrates previously colonized by other fungi and not only scale insects [4,5,8,12]. This broadens the taxonomic concept of *Podonectria,* which is further supported by molecular analyses in this study.

*Podonectria novae-zelandiae* Dingley, Trans. & Proc. Roy. Soc. N.Z. 81: 496 (1954) (Figure 3 and Figure 4).

MycoBank number: MB 304079

Habitat associated with scale insects *Kuwanaspis howardi* on *Phyllostachys heteroclada*. *Sexual morph*: *Stromata* byssoid, brown, well-developed, and covered the scale insects. *Ascomata* solitary, rarely aggregated, superficial on the byssoid stroma, concomitant with sporodochia, light yellow, covered with long hairs, 150–415 μm high ($\bar{x}$ = 240 μm, *n* = 20), 100–350 μm wide (without hairs) ($\bar{x}$ = 192 μm, *n* = 30). *Hairs* 60–280 μm long, multiseptate, 3–6.5 μm wide, straight, or curved, abundant, hyaline, slightly narrowed toward the apex, 1–2.5 μm thick-walled (*n* = 30). *Peridium* 60–100 μm thick, usually wider at the base, composed of hyaline suborbicular cells forming *textura angularis*, the cells measuring 5.5–12 × 4.5–10 μm ($\bar{x}$ = 8.9 × 7.0 μm, *n* = 20). *Hamathecium* 1.5–3 μm diameter ($\bar{x}$ = 2.3 μm, *n* = 30), 1 μm diameter at the apex, longer than the asci, numerous, filiform, curved, septate, branched pseudoparaphyses. *Asci* 220–340 × 18–26 μm ($\bar{x}$ = 267 × 21 μm, *n* = 20), 8-spored, bitunicate, cylindrical, straight, or curved, rounded at apex. *Ascospores* 100–160 × 7–10 μm ($\bar{x}$ = 138 × 9 μm, *n* = 30), fasciculate, parallel, long-clavate, rounded at ends, multiseptate, 10–22 septa with slight constriction, curved, hyaline, smooth. *Asexual morph*: *Stromata* hard, white to grey-brown, cushion-shaped, formed directly on host scales with 1–4 sporodochia. *Sporodochia* erupted, white, yellowish to grey-brown, scattered or aggregated. *Conidiophores* inconspicuous, short, 1–2 celled, the cells 3–7 × 4–10 μm ($\bar{x}$ = 5.0 × 7.5 μm, *n* = 30), usually globose, subglobose, or shortly cylindrical, attached with 1–2 conidiogenous cells. *Conidiogenous cells* 3–7 × 4–11 μm ($\bar{x}$ = 7.3 × 6.5 μm, *n* = 30), globose or ellipsoidal. *Conidia* usually with two and three “arms”, occasionally one and four “arms”, each “arm” varies in length and slightly divergent, 85–163 μm long ($\bar{x}$ = 117 μm, *n* = 70), 7–11 μm wide ($\bar{x}$ = 9 μm, *n* = 70) with 11–25 septa, mature conidium tapering toward the acute apex. All “arms” of single conidium joined at a basal oval or irregular cell, measuring 4–7 × 5–10 μm ($\bar{x}$ = 5.3 × 7.1 μm, *n* = 40).

Material examined: CHINA, Sichuan Province, Ya’an City, Lushan County (102°55′58.13″ E, 30°15′24.07″ N, Alt. 1116 m), on scale insect *Kuwanaspis howardi*, 10 June 2020, Xiu-lan Xu, XXL202006006 (SICAU 21-0005), living culture SICAUCC 21-0005; ibid. XXL202006005 (SICAU 21-0004), living culture SICAUCC 21-0004.

Culture characters: Conidia germinate on PDA within 12 h, and the cultures grow slowly on PDA. Colonies reach 2 cm in diameter after 25 days. Colonies from single conidia are flocculent and hard, with irregular margins. The mycelium is creamy white to light lemon yellow starting at the center but gradually becoming brown to dark brown after 20 days. Aerial hyphae cluster and raise straightly, measuring 2–3 μm diam. Conidia develop on small, sparsely distributed mycelial clumps after two months. Conidiophores moniliform, branched, multi-celled, and longer than those in nature. Conidia commonly have three “arms”, occasionally two and four “arms”, rarely one and five “arms”, each “arm” with 17–22 septa, measuring 115–145 μm long, 6.5–10 μm wide ($\bar{x}$ = 128 × 7.9 μm, *n* = 30). Ascospores germinate on PDA within 12 h, and the cultures grow slowly on PDA. Colonies reach 1 cm in diameter after 20 days. Colonies from single ascospores are cottony, with regular margin; the mycelium is creamy white to yellow; and the back of colonies is brown, with concentric rings.

Notes: Here, we follow the recommendation of Rossman et al. [62] by adopting *Podonectria* over *Tetracrium*. The asexual morph of *P. novae-zelandiae* was reported by Dao et al. [5], and was supported with morphology and molecular data. Our observations agree with the descriptions provided by Rossman [4] and Dao et al. [5]. Nucleotide comparison of ITS and LSU (SICAUCC 21-0005) reveals high similarity to *P. novae-zelandiae* (isolate PUcS13, similarities = 473/476 (99%), 0 gaps (0%); similarities = 517/518 (99%), 0 gaps (0%), respectively) in Dao et al. [5]; however, the latter lack SSU, *tef1-α*, and *rpb2* sequences for further comparisons. The conidia produced here in culture were similar to those on scale insects in the field.

*Podonectria kuwanaspidis* X.L. Xu & C.L. Yang sp. nov. (Figure 5)

MycoBank number: MB 838465

*Etymology*: In reference to the generic name for the associated scale insect.

Holotype: SICAU 21-0002.

Habitat associated with scale insects *Kuwanaspis howardi* on *Phyllostachys heteroclada*. *Sexual morph*: *Stromata* byssoid, white or brown, well-developed and covering the scale insects, or forming a thin, white, and byssoid layer, which spreads out from the scale over the stem. *Ascomata* solitary to aggregated, superficial on byssoid stroma around the scale hosts, or extending far beyond the scale byssoid stroma, globose to subglobose, creamy white to dirty white, covered with hairs, 200–590 μm high ($\bar{x}$ = 424 μm, *n* = 20), 140–600 μm wide ($\bar{x}$ = 346 μm, *n* = 70). *Hairs* 30–120 μm long, 0–4 septa, 8–16 μm wide at the base, 2–7 μm wide at the apex, abundant, hyaline, distinctly narrowed toward the apex, 2–4.5 μm thick-walled (*n* = 40). *Ostiole* 40–100 μm wide. *Peridium* 40–170 μm thick ($\bar{x}$ = 72 μm, *n* = 30), usually wider at the base, composed of hyaline elongated cells forming *textura prismatica* to *textura angularis*, becoming globose toward outside, the cells measuring 6.5–15× 8–20 μm ($\bar{x}$ = 10 × 14 μm, *n* = 30) μm. *Hamathecium* 1.5–3.5 μm diameter ($\bar{x}$ = 2.2 μm, *n* = 40) at the base, 1 μm diameter at the apex, longer than the asci, numerous, filiform, curved, septate, branched pseudoparaphyses. *Asci* 185–250 × 15–25 μm ($\bar{x}$ = 219 × 20 μm, *n* = 40), 8-spored, bitunicate, cylindrical, straight or curved, rounded at apex, shortly pedicellate. *Ascospores* 150–240 × 5–7 μm ($\bar{x}$ = 199 × 5.8 μm, *n* = 40), fasciculate, parallel or spiral, broadly filiform, cylindrical to long fusiform, elongate, rounded at ends, multiseptate, 16–31 septa without constriction, usually tapering toward the base, curved, hyaline, smooth. *Asexual morph*: Undetermined.

Material examined: CHINA, Sichuan Province, Ya’an City, Lushan County (102°55′58.13″ E, 30°15′24.07″ N, Alt. 1116 m), on scale insect *Kuwanaspis howardi*, 10 June 2020, Xiu-lan Xu, XXL202006002 (SICAU 21-0002, holotype), ex-type culture, SICAUCC 21-0002; ibid. XXL202006003 (SICAU 21-0003, paratype), living culture SICAUCC 21-0003. ibid. Yucheng District, Kongping Township (103°2′59.87″ E, 29°50′8.56″ N, Alt. 1133 m), on scale insect *Kuwanaspis howardi*, 19 September 2018, Xiu-lan Xu, YCL201810014 (SICAU 21-0007, paratype), living culture SICAUCC 21-0007.

Culture characters: Ascospores germinating on PDA within 12 h, and the cultures grow slowly on PDA. Colonies reach 2 cm in diameter after 25 days. Colonies from single ascospores are cottony, cling to the medium, with regular margin; the mycelium is creamy white to pale yellow but gradually becomes pale brown after 30 days.

Notes: This new taxon resembles species of *Podonectria*, in having superficial, bright, or lightly colored fruiting bodies and hairs obscuring the outer wall of ascoma. Morphologically *Podonectria kuwanaspidis* is comparable with *P. novae-zelandiae*. It has shorter (30–120 vs. 60–280 μm) and thicker-walled hairs (2–4.5 vs.1–2.5 μm), longer and narrower ascospores (150–240 × 5–7 vs. 100–160 × 7–10 μm). The ITS base-pair comparison between *Podonectria kuwanaspidis* (SICAUCC 21-0002) and phylogenetically affiliated *P. sichuanensis* (SICAU 16-0003) reveals 15% (including 20 gaps, 4%) nucleotide differences; the nucleotide differences in the SSU, LSU, *tef1-α*, and *rpb2* region between them are 1% (0 gaps, 0%), 3% (5 gaps, 0%), 4% (0 gaps, 0%), and 10% (0 gaps, 0%), respectively. Hence, we describe our collection as a new species in *Podonectria*, as recommended by Jeewon and Hyde [63].

*Nectriaceae* Tul. & C. Tul., Select. fung. carpol. (Paris) 3:3 (1865)

*Microcera* Desm., Annls Sci. Nat., Bot., sér. 3 10:359 (1848)

*Microcera kuwanaspidis* X.L. Xu & C.L. Yang sp. nov. (Figure 6)

MycoBank number: MB 838464

*Etymology*: In reference to the generic name for the associated scale insect.

Holotype: SICAU 21-0006.

Habitat associated with scale insects *Kuwanaspis howardi* on bamboo. *Sexual morph*: Undetermined. *Asexual morph*: *Stromata* 500–690 μm long, 410–600 μm wide ($\bar{x}$ = 614 × 524 μm, *n* = 20), ellipsoid, orange-red, completely covering a single scale insect, or absent. *Sporodochia* 190–280 μm long, 150–300 μm wide ($\bar{x}$ = 240 × 227 μm, *n* = 20), formed singly on the margin of the stroma, or rarely in groups of one to three on the margin of the scale covers. *Conidiophores* with developing macroconidia form a pink upright mass. *Macroconidia* (80–)95–120 μm long × 6.5–8.5 ($\bar{x}$ = 107 × 7.3 μm, *n* = 20) μm wide, hyaline, cylindrical, slightly curved, slender toward each end, 3–8 septate, mostly 5–6–7 septate, difficult to distinguish apical cell and basal cell. *Microconidia* and *chlamydospores* were not observed.

Material examined: CHINA, Sichuan Province, Ya’an City, Lushan County (102°55′58.13″ E, 30°15′24.07″ N, Alt. 1116 m), on scale insect *Kuwanaspis howardi* on *Phyllostachys heteroclada*, 10 June 2020, Xiu-lan Xu, XXL202006007 (SICAU 21-0006, holotype), ex-type culture SICAUCC 21-0006, additional GenBank Number: SSU = MW462896; CHINA, Sichuan Province, Meishan City, Hongya County (103°14′2.64″ E, 29°41′53.07″ N, Alt. 538 m), on scale insect *Kuwanaspis howardi* on *Pleioblastus amarus*, 9 March 2021, Chun-lin Yang, YCL202103001 (SICAU 21-0009, paratype), living culture SICAUCC 21-0009, additional GenBank Number: SSU = MZ029435.

Culture characters: Colonies from a single macroconidium on PDA grow slowly and reach approximately 2.2 cm in diameter after 12 days at 25 °C, circular, flat, whitish to bright orange with white mycelium on the surface forming concentric circles, and the back of colonies is bright orange.

Notes: Distinguished from the red-headed fungus *Microcera coccophila* [18,64], in which the sporodochium is usually formed in groups on margin of dead scale or their covers accompanied with perithecia surround the edge of scale covers. However, this new species has distinct stroma covering the host, with a single sporodochium at the edge and without perithecia being discovered. Furthermore, although they are similar in size, *Microcera kuwanaspidis* is different from *M. coccophila* in numbers of septa (3–8 vs. 7–9). *Microcera kuwanaspidis* clusters with *M. coccophila* (CBS 310.34) with 100% ML and 1.00 BYPP support; however, striking base-pair differences are noted, viz. 1% (0 gaps, 0%), 1% (0 gaps, 0%), 1% (0 gaps, 0%), 13% (23 gaps, 4%), 4% (0 gaps, 0%), 3% (0 gaps, 0%), 5% (3 gaps, 0%), and 9% (6 gaps, 1%) in the ITS, LSU, *rpb2*, *tef1-α*, *acl1*, *act*, *cmdA*, and *his3* DNA sequence data, respectively. According to the guidelines of Jeewon and Hyde [63], our collection is proposed as a new species.

## 4. Discussion

Mycologists have questioned the exact familial placement of *Podonectria* since the beginning of its establishment. Dingley [3] placed the genus in Clavicipitaceae (Hypocreales). Rossman transferred it into the Pleosporaceae (Pleosporales) due to its bitunicate asci rather than the unitunicate asci found in Hypocreales [4,7]. Barr transferred *Podonectria* to Tubeufiaceae [65], which was erected [66] to accommodate pleosporaceous taxa that are typically hyper saprobic on other fungi or substrates previously colonized by other fungi, hyperparasitic on foliicolous fungi, parasitic on scale insects, or occasionally parasitic on living leaves. This treatment was followed by subsequent authors [9,67,68,69]. However, Tubeufiaceae, which was comprehensively reviewed by Boonmee et al. [70], was accommodated in a new order, Tubeufiales [58]. This placement was followed by Wijayawardene et al. [71,72] and Hongsanan et al. [73]. However, Dao et al. [5] proposed Podonectriaceae, a new family in Pleosporales, to accommodate this genus, which was confirmed by ITS and LSU data. This placement was supported by Yang et al. [12], in which *Podonectria sichuanensis* was identified based on morphological characteristics and phylogenetic analyses. Based on the phylogenetic results of combined ITS, LSU, SSU, *tef1-α,* and *rpb2* data in this current study, we confirm Podonectriaceae as an accepted family in the suborder Pleosporineae [49]. Podonectriaceae is phylogenetically closely related to Pseudopyrenochaetaceae that has been established to accommodate two species, viz. *Pseudopyrenochaeta lycopersici* and *P. terrestris* [57]. However, the two families are morphologically distinct. Pseudopyrenochaetaceae has pycnidial conidiomata, filiform conidiophores, and aseptate, cylindrical to allantoid conidia, whereas Podonectriaceae comprises sporodochial conidiomata, moniliform or inconspicuous conidiophores, and 1–4 armed, multiseptated conidia. In addition, the coelomycete genera *Tetranacrium* that has septate tetraradiate conidia [74,75] was documented as the anamorph associated with *Podonectria gahnia* according to substrate observation [4]. However, the association is somewhat confused, as it lacks further phylogenetic investigations and taxonomic studies. Identical molecular sequences of *Podonectria novae-zelandiae* in our study confirmed the link between the sexual morphs and asexual morphs in *Tetracrium*. *Podonectria* was reported to be associated with scale insects on various hosts in previous studies [1,4,5,58]. In this paper, we isolated *Podonectria sichuanensis* (SICAUCC 21-0001) on the ascomata of *Neostagonosporella sichuanensis* in our sampling site and confirm that the *Podonectria* species are not only parasitic on scale insects but also on other fungi or substrates previously colonized by other fungi [12]. According to published studies, most species of *Podonectria* are associated with armored scale insects, in addition to being associated with the mostly reported hosts *Citrus aurantium* L. and *C. nobilis* Lour. (Rutaceae) [1,4] and the known host plants associated with *Podonectria* are *Calluna vulgaris* Salisb. (Ericaceae), *Gahnia setifolia* (A. Rich) Hook.f., *G. xanthocarpa* (Hook.f.) Hook. f. (Cyperaceae), *Juniperus bermudiana* L. (Cupressaceae), *Olearia rani* Druce (Asteraceae), *Phyllostachys heteroclada* (Poaceae) and *Podocarpus ferrugineus* G. Benn. ex D. Don (Podocarpaceae) [3,4,5,12].

Gräfenhan et al. [18] reported an association of *Microcera* to *Fusarium*, *Cladosterigma* Pat., *Mycogloea* L.S. Olive, *Tetracrium* Henn., and accepted four species in *Microcera*. Nowadays, taxonomic concepts based on multi-gene phylogenetic inference have provided a deeper understanding of phylogenetic relationships than those based on individual gene regions [76,77,78,79]. Recently, combined ITS-LSU-*tef1-α*-*acl1*-*act*-*cmdA*-*his3*-*rpb1*-*rpb2*-*tub2* datasets were used to clarify intraspecific and intergeneric relationships within Nectriaceae [19], and combined ITS-LSU-*tef1-α*-*cmdA*-*rpb2*-*tub2* datasets were similarly used for Hypocreales [80]. In this paper, *Microcera kuwanaspidis* can be distinguished from *M. coccophila* and is established as new species on account of base-pair differences, especially in the *tef1-α* (13%), *acl1* (4%), *act* (3%), *cmdA* (5%), and *his3* (9%). The *Microcera* species have been mostly reported associated with armored scale insects on citrus (Rutaceae), viz. *Aonidiella aurantii*, *A. citrina*, *Lepidosaphes beckii*, *Unaspis citri*, and *Quadraspidiotus perniciosus* on *Pyrus communis*, *Prunus domestica* and *P. cerasus* (Rosaceae), as well associated with nut scale *Eulecanium tiliae* (Hemiptera: Coccidae) on *Salix* sp. (Salicaceae) and *Fraxinus excelsior* (Oleaceae), and an unknown scale insect on *Broussonetia kazinoki* × *B. papyrifera* (Moraceae), *Laurus nobilis* (Lauraceae), *Citrus maxima* (Rutaceae), and apple trees [18,60,64].

In China, the entomopathogenic fungi associated with scale insects was mainly focused on commercial *Citrus* plants in the 1990s. *Verticillium lecanii* (Zimm) Viegas is the most common fungus that is parasitic on scale insects on *Citrus* since its discovery from Guizhou Province in 1982 [81]. Subsequently, *Aschersonia duplex* Berk., *Beauveria bassiana* (Bals.-Criv.) Vuill., *Fusarium juruanum* Henn., *F. moniliforme* Sheld., *Microcera coccophila*, *Nigrospora sphaerica* (Sacc) Mason, and *Podonectria coccicola* have also been reported to be associated with the scale insects on citrus [82,83,84]. *Microcera* and *Podonectria* were commonly encountered on scale insects within tree canopies and occurred throughout the year but were more noticeable under wet and humid conditions [5,64,85,86], consistent with the observations in this study. Presently, *Microcera coccophila* and *Podonectria coccicola* have been the most commonly and worldwide recorded species on scale insects, especially on orange trees [1,4,7,85,86,87,88,89]. This paper provides new records for three entomopathogenic fungi, *Podonectria kuwanaspidis*, *P. novae-zelandiae*, and *Microcera kuwanaspidis* on armored insect scale from bamboo in China. According to the field observation from 2015 to 2020, the three species are commonly associated with *Kuwanaspis howardi* on native bamboo, especially on *Phyllostachys heteroclada*, and they effectively cause the scale insect hosts to be infected, which ultimately results in death. As documented by Rossman [4] and Dao et al. [64], the role of entomopathogens in the biological control of destructive scale insects on citrus trees was usually controlled by chemical sprays. These entomopathogenic fungi should be further screened to assess their potential for commercial development as biological control agents.

## Figures and Tables

**Figure 1 jof-07-00628-f001:**
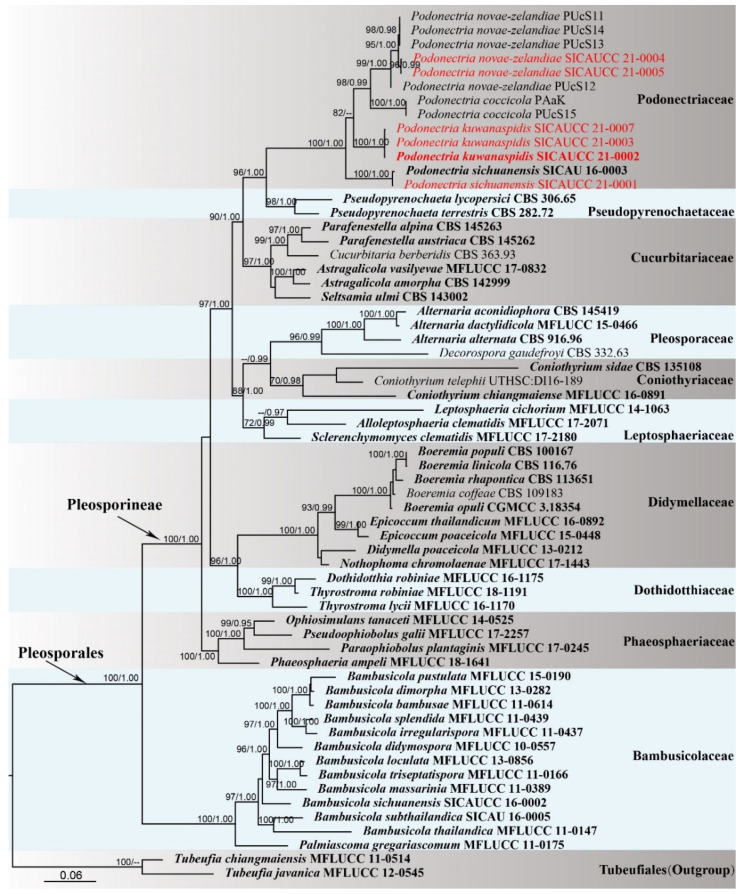
Phylogram generated from RAxML analysis based on ITS, LSU, SSU, *tef1-α*, and *rpb2* Scheme 70. and Bayesian posterior probabilities (BYPP, right) equal to or greater than 0.95 are indicated at the nodes respectively. The sequences from ex-type strains are in bold. The newly generated sequence is in red.

**Figure 2 jof-07-00628-f002:**
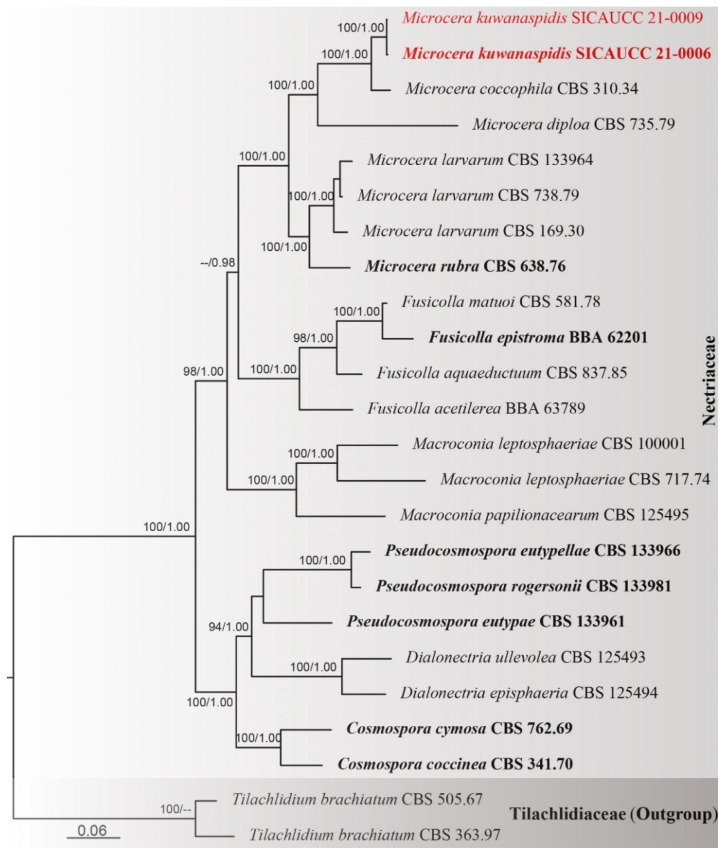
Phylogram generated from RAxML analysis based on combined ITS, LSU, *tef1-α*, *rpb1*, *rpb2*, *acl1*, *act*, *tub2*, *cmdA*, and *his3* sequence data of *Microcera* isolates. Bootstrap support values for maximum likelihood (ML, left) higher than 70% and Bayesian posterior probabilities (BYPP, right) equal to or greater than 0.95 are indicated at the nodes respectively. The sequences from ex-type strains are in bold. The newly generated sequence is in red.

**Figure 3 jof-07-00628-f003:**
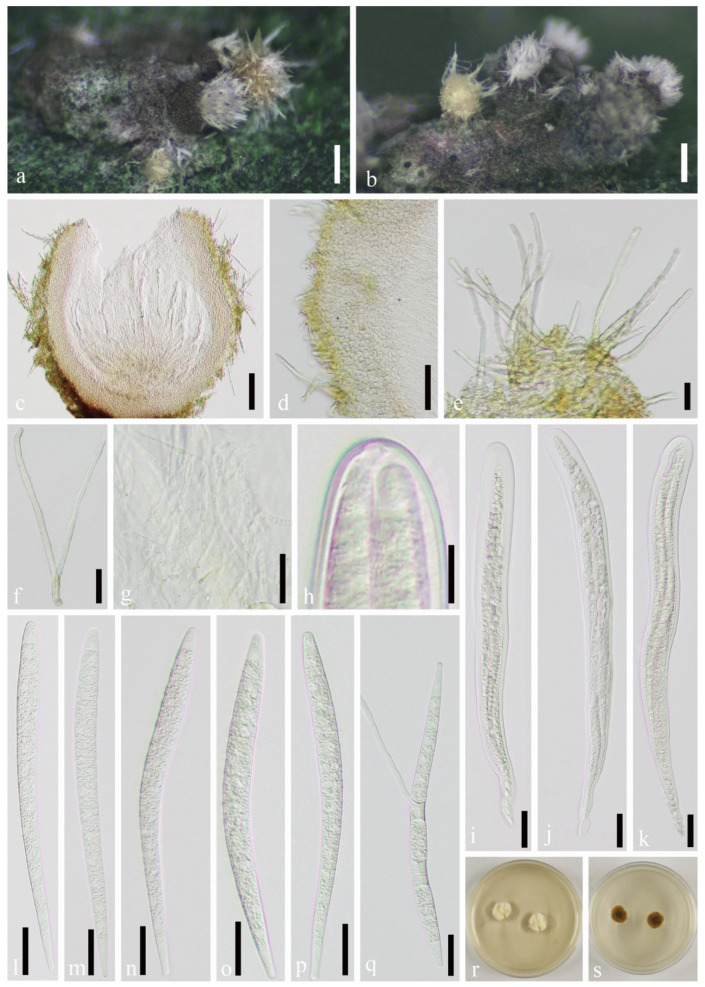
*Podonectria novae-zelandiae* (SICAU 21-0005). (**a**,**b**) Ascomata and sporodochia on host substrate. (**c**) Section through ascoma. (**d**) Peridium. (**e**,**f**) Hairs covering on ascoma. (**g**) Pseudoparaphyses. (**h**) Ocular chamber. (**i**–**k**) Asci. (**l**–**p**) Ascospores. (**q**) Germinated ascospores. (**r**,**s**) Colonies on PDA after 18 days. Scale bars: (**a**,**b**) 200 μm, (**c**) 100 μm, (**d**) 50 μm, (**e**–**g**) 20 μm, (**h**) 10 μm, (**i**–**q**) 20 μm.

**Figure 4 jof-07-00628-f004:**
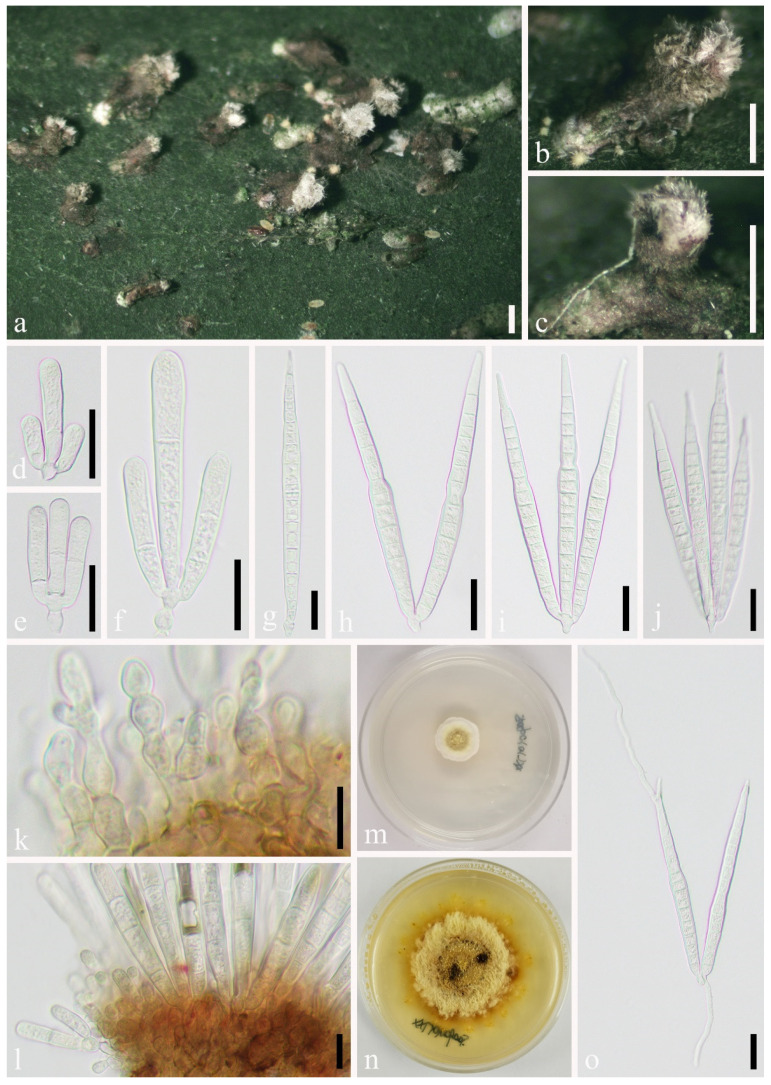
*Podonectria novae-zelandiae* (SICAU 21-0004). (**a**–**c**) Sporodochia and ascomata on host substrate. (**d**–**f**) Immature conidia. (**g**–**j**) Mature conidia. (**k**,**l**) Conidiophores. (**m**,**n**) Colonies on PDA after 20 days and 60 days. (**o**) Germinated conidium. Scale bars: (**a**–**c**) 500 μm, (**d**–**k**) 20 μm, (**l**,**o**) 10 μm.

**Figure 5 jof-07-00628-f005:**
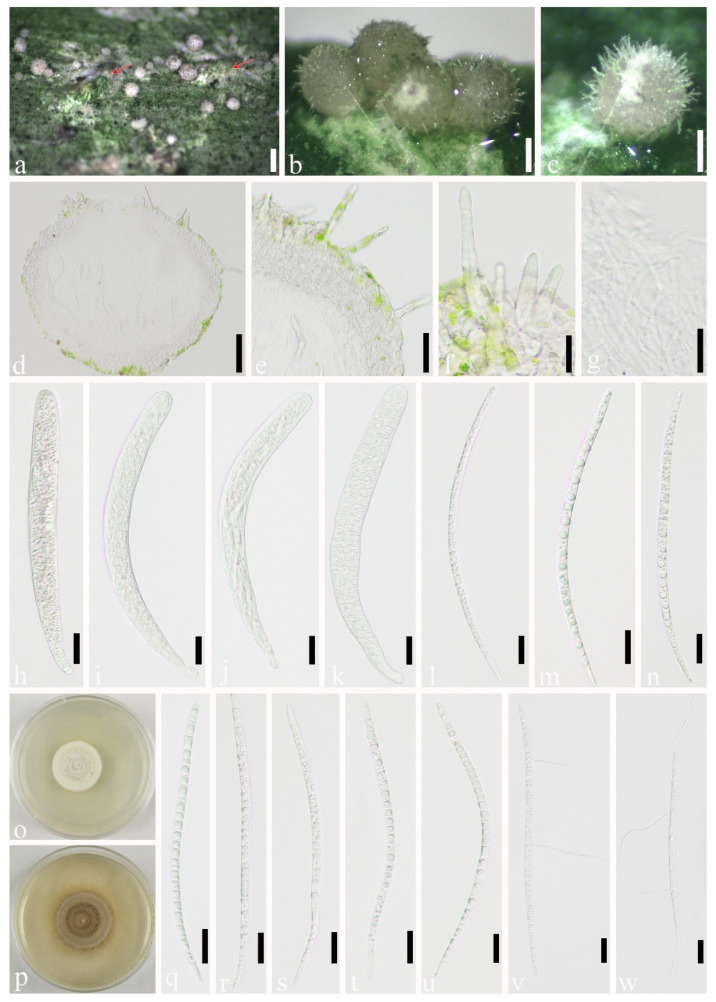
*Podonectria kuwanaspidis* (SICAU 21-0002, holotype). (**a**) Ascomata on or around the scalehost (red arrow). (**b**) Aggregated ascomata. (**c**) Solitary ascomata. (**d**) Section through ascoma. (**e**) Peridium. (**f**) Hairs covering ascoma. (**g**) Pseudoparaphyses. (**h**–**k**) Asci. (**l**–**n**,**q**–**u**) Ascospores. (**o**,**p**) Colonies on PDA after 25 days and 50 days. (**v**,**w**) Germinated ascospores. Scale bars: (**a**) 500 μm, (**b**,**c**) 200 μm, (**d**) 100 μm, (**e**) 50 μm, (**f**–**n**) 20 μm, (**q**–**w**) 20 μm.

**Figure 6 jof-07-00628-f006:**
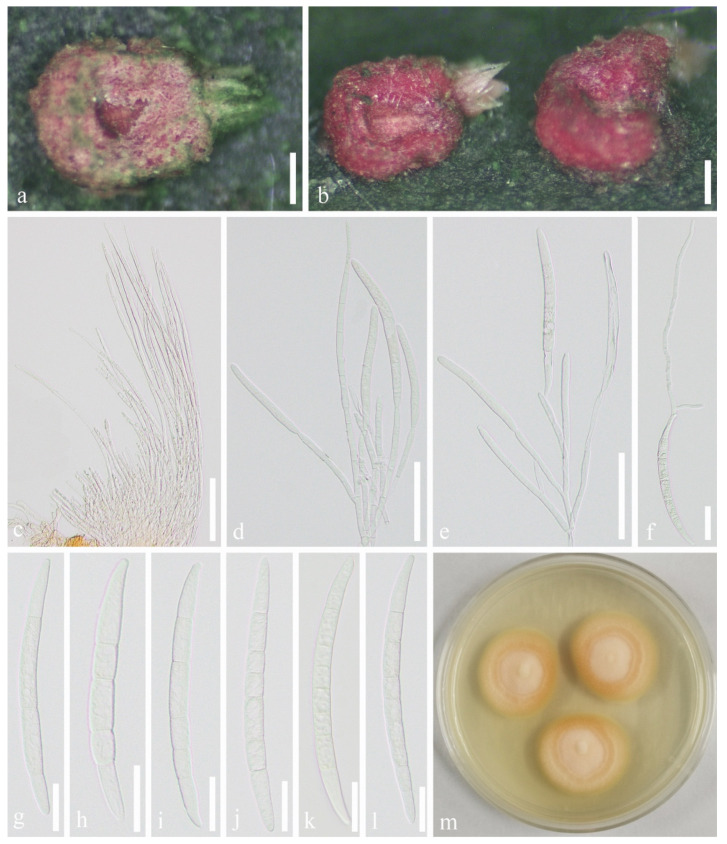
*Microcera kuwanaspidis* (SICAU 21-0006, holotype). (**a**,**b**) Stromata and sporodochia on host substrate. (**c**–**e**) Conidiophore with developing macroconidia. (**f**) Germinated conidium. (**g**–**l**) Macroconidia. (**m**) Colonies on PDA after 12 days. Scale bars: (**a**,**b**) 200 μm, (**c**–**e**) 50 μm, (**f**–**l**) 20 μm.

**Table 1 jof-07-00628-t001:** GenBank accession numbers of strains in Pleosporales and Tubeufiales used for the phylogenetic analyses of *Podonectria*.

Species	Strain/Voucher No.	GenBank Accession Numbers					References
		ITS	LSU	SSU	*tef1-α*	*rpb2*	
*Alternaria alternata*	CBS 916.96 ^T^	AF347031	DQ678082	KC584507	KC584634	KC584375	[35]
*Alternaria aconidiophora*	CBS 145419 ^T^	LR133931	–	–	LR133968	LR133967	[36]
*Alternaria dactylidicola*	MFLUCC 15-0466 ^T^	KY703616	KY703617	KY703618	–	KY750720	[37]
*Alloleptosphaeria clematidis*	MFLUCC 17-2071 ^T^	MT310604	MT214557	MT226674	MT394736	MT394685	[38]
*Astragalicola amorpha*	CBS 142999 ^T^	MF795753	MF795753	–	MF795842	MF795795	[39]
*Astragalicola vasilyevae*	MFLUCC 17-0832 ^T^	MG828870	MG828986	MG829098	MG829193	MG829248	[40]
*Bambusicola bambusae*	MFLUCC 11-0614 ^T^	JX442031	JX442035	JX442039	KP761722	KP761718	[41,42]
*Bambusicola irregularispora*	MFLUCC 11-0437 ^T^	JX442032	JX442036	JX442040	KP761723	KP761719	[41,42]
*Bambusicola massarinia*	MFLUCC 11-0389 ^T^	JX442033	JX442037	JX442041	KP761725	KP761716	[41,42]
*Bambusicola splendida*	MFLUCC 11-0439 ^T^	JX442034	JX442038	JX442042	KP761726	KP761717	[41,42]
*Bambusicola didymospora*	MFLUCC 10-0557 ^T^	KU940116	KU863105	KU872110	KU940188	KU940163	[30]
*Bambusicola pustulata*	MFLUCC 15-0190 ^T^	KU940118	KU863107	KU872112	KU940190	KU940165	[30]
*Bambusicola thailandica*	MFLUCC 11-0147 ^T^	KU940119	KU863108	KU872113	KU940191	KU940166	[30]
*Bambusicola triseptatispora*	MFLUCC 11-0166 ^T^	KU940120	KU863109	–	–	KU940167	[30]
*Bambusicola dimorpha*	MFLUCC 13-0282 ^T^	KY026582	KY000661	KY038354	–	KY056663	[37]
*Bambusicola loculata*	MFLUCC 13-0856 ^T^	KP761732	KP761729	KP761735	KP761724	KP761715	[42]
*Bambusicola sichuanensis*	SICAUCC 16-0002 ^T^	MK253473	MK253532	MK253528	MK262828	MK262830	[43]
*Bambusicola subthailandica*	SICAU 16-0005 ^T^	MK253474	MK253533	MK253529	MK262829	MK262831	[43]
*Boeremia coffeae*	CBS 109183	GU237748	GU237943	–	KY484678	KT389566	[44]
*Boeremia rhapontica*	CBS 113651 ^T^	KY484662	–	–	KY484713	–	[44]
*Boeremia opuli*	CGMCC 3.18354 ^T^	KY742045	KY742199	–	–	KY742133	[44]
*Boeremia linicola*	CBS 116.76 ^T^	GU237754	GU237938	–	KY484705	KT389574	[44]
*Boeremia populi*	CBS 100167 ^T^	GU237707	GU237939	–	KY484706	–	[44]
*Coniothyrium telephii*	UTHSC DI16-189	LT796830	LN907332	–	–	LT796990	[45]
*Coniothyrium chiangmaiense*	MFLUCC 16-0891 ^T^	KY568987	KY550384	KY550385	–	KY607015	[37]
*Coniothyrium sidae*	CBS 135108 ^T^	KF251149	KF251653	–	KF253109	KF252158	[46]
*Cucurbitaria berberidis*	CBS 363.93	JF740191	GQ387606	–	–	–	[47]
*Decorospora gaudefroyi*	CBS 332.63	AF394541	–	AF394542	–	–	[48]
*Didymella poaceicola*	MFLUCC 13-0212 ^T^	KX965726	KX954395	–	–	KX898364	[37]
*Dothidotthia robiniae*	MFLUCC 16-1175 ^T^	MK751727	MK751817	MK751762	MK908017	MK920237	[49]
*Epicoccum thailandicum*	MFLUCC 16-0892 ^T^	KY703619	KY703620	–	–	–	[37]
*Epicoccum poaceicola*	MFLUCC 15-0448 ^T^	KX965727	KX954396	–	–	KX898365	[37]
*Leptosphaeria cichorium*	MFLUCC 14-1063 ^T^	KT454720	KT454712	KT454728	–	–	[50]
*Nothophoma chromolaenae*	MFLUCC 17-1443 ^T^	MT214364	MT214458	MT214410	–	–	[51]
*Ophiosimulans tanaceti*	MFLUCC 14-0525 ^T^	KU738890	KU738891	KU738892	MG520910	–	[52,53]
*Palmiascoma gregariascomum*	MFLUCC 11-0175 ^T^	KP744452	KP744495	KP753958	–	KP998466	[54]
*Parafenestella austriaca*	CBS 145262 ^T^	MK356304	MK356304	–	MK357576	MK357532	[55]
*Parafenestella alpina*	CBS 145263 ^T^	MK356302	MK356302	–	MK357574	MK357530	[55]
*Paraophiobolus plantaginis*	MFLUCC 17-0245 ^T^	KY797641	KY815010	KY815012	MG520913	–	[53]
*Phaeosphaeria ampeli*	MFLUCC 18-1641 ^T^	MK503797	MK503808	MK503814	MK503802	–	[56]
*Podonectria coccicola*	DAR 81026	KU587798	KU519419	–	–	–	[5]
*Podonectria coccicola*	PUcS15	KU720533	KU519420	–	–	–	[5]
*Podonectria novae-zelandiae*	PUcS14	KU720535	KU559551	–	–	–	[5]
*Podonectria novae-zelandiae*	PUcS13	KU720538	KU559548	–	–	–	[5]
*Podonectria novae-zelandiae*	PUcS12	KU720537	KU529802	–	–	–	[5]
*Podonectria novae-zelandiae*	PUcS11	KU720536	KU568479	–	–	–	[5]
*Podonectria sichuanensis*	SICAU 16-0003 ^T^	MK305903	MK296471	MK296467	MK313852	MK313855	[12]
***Podonectria sichuanensis***	**SICAUCC 21-0001**	**MW484988**	**MW462899**	**MW462891**	**MW462111**	**MW462118**	**This study**
***Podonectria kuwanaspidis***	**SICAUCC 21-0002 ^T^**	**MW484989**	**MW462900**	**MW462892**	**MW462112**	**MW462119**	**This study**
***Podonectria kuwanaspidis***	**SICAUCC 21-0003**	**MW484990**	**MW462901**	**MW462893**	**MW462113**	**MW462120**	**This study**
***Podonectria novae-zelandiae***	**SICAUCC 21-0004**	**MW484991**	**MW462902**	**MW462894**	**MW462114**	**MW462121**	**This study**
***Podonectria novae-zelandiae***	**SICAUCC 21-0005**	**MW484992**	**MW462903**	**MW462895**	**MW462115**	**MW462122**	**This study**
***Podonectria kuwanaspidis***	**SICAUCC 21-0007**	**MW484994**	**MW462905**	**MW462897**	**MW462116**	**MW462123**	**This study**
*Pseudoophiobolus galii*	MFLUCC 17-2257 ^T^	MG520947	MG520967	MG520989	MG520926	–	[53]
*Pseudopyrenochaeta lycopersici*	CBS 306.65 ^T^	AY649587	EU754205	–	–	LT717680	[57]
*Pseudopyrenochaeta terrestris*	CBS 282.72 ^T^	LT623228	LT623216	–	–	LT623287	[57]
*Sclerenchymomyces clematidis*	MFLUCC 17-2180 ^T^	MT310605	MT214558	MT226675	MT394737	MT394686	[38]
*Seltsamia ulmi*	CBS 143002 ^T^	MF795794	MF795794	MF795794	MF795882	MF795836	[39]
*Thyrostroma lycii*	MFLUCC 16-1170 ^T^	MK751734	MK751824	MK751769	MK908024	MK920241	[49]
*Thyrostroma robiniae*	MFLUCC 18-1191 ^T^	MK751735	MK751825	MK751770	MK908025	MK920242	[49]
*Tubeufia javanica*	MFLUCC 12-0545 ^T^	KJ880034	KJ880036	KJ880035	KJ880037	–	[58]
*Tubeufia chiangmaiensis*	MFLUCC 11-0514 ^T^	KF301530	KF301538	KF301543	KF301557	–	[58]

Notes: The superscript T represents ex-type or ex-epitype isolates. “–” means that the sequence is missing or unavailable. New sequences are listed in bold. Abbreviations. CBS: Centraalbureau voor Schimmelcultures, Utrecht, Netherlands; CGMCC: China General Microbiological Culture Collection Center; DAR: New South Wales Plant Pathology Herbarium, Orange Agricultural Institute, Orange, NSW, Australia; MFLUCC: Mae Fah Luang University Culture Collection, Chiang Rai, Thailand; PUcS: unspecified; UTHSC: Fungus Testing Laboratory of the University of Texas Health Science Center at San Antonio, San Antonio, TX, USA; SICAUCC: Sichuan Agricultural University Culture Collection, Sichuan, China; SICAU: Herbarium of Sichuan Agricultural University, Sichuan, China.

**Table 2 jof-07-00628-t002:** GenBank accession numbers of strains in Nectriaceae used for the phylogenetic analyses of *Microcera*.

Species	Strain/Voucher No.	GenBank Accession No.										References
		*acl1*	*act*	*cmdA*	*his3*	ITS	LSU	*rpb1*	*rpb2*	*tef1-α*	*tub2*	
*Cosmospora coccinea*	CBS 341.70 ^T^	HQ897913	KM231221	KM231398	KM231550	HQ897827	KM231692	KM232242	HQ897777	KM231947	KM232086	[18,19]
*Cosmospora cymosa*	CBS 762.69 ^T^	HQ897914	KM231222	KM231399	KM231551	HQ897828	KM231693	KM232243	HQ897778	KM231948	KM232087	[18,19]
*Dialonectria episphaeria*	CBS 125494 = TG 2006-11	HQ897892	KM231227	KM231404	KM231556	HQ897811	KM231697	KM232248	HQ897756	KM231953	KM232092	[18,19]
*Dialonectria ullevolea*	CBS 125493 = TG 2007-56	HQ897918	KM231226	KM231403	KM231555	KM231821	KM231696	KM232247	HQ897782	KM231952	KM232091	[18,19]
*Fusicolla acetilerea*	BBA 63789 ^T^ = IMI 181488 = NRRL20827	KM231065	–	–	–	HQ897790	U88108	–	HQ897701	–	–	[18]
*Fusicolla aquaeductuum*	CBS 837.85 = BBA 64559 = NRRL 20865	KM231067	-	KM231406	-	KM231823	KM231699	KM232250	HQ897744	KM231955	KM232094	[19]
*Fusicolla epistroma*	BBA 62201 ^T^ = IMI 85601 = NRRL 20439	KM231069	–	–	–	–	AF228352	–	HQ897765	–	–	[18]
*Fusicolla matuoi*	CBS 581.78 = ATCC 18694 = MAFF 238445 = NRRL 20427	KM231070	KM231228	KM231405	KM231557	KM231822	KM231698	KM232249	HQ897720	KM231954	KM232093	[18,19]
*Macroconia papilionacearum*	CBS 125495	HQ897912	KM231233	KM231411	KM231561	HQ897826	KM231704	KM232254	HQ897776	KM231958	KM232096	[18,19]
*Macroconia leptosphaeriae*	CBS 717.74	KM231062	KM231236	KM231414	KM231564	KM231827	KM231707	KM232257	KM232390	JF735695	KM232099	[18,19]
*Macroconia leptosphaeriae*	CBS 100001 = CBS H-6030	KM231063	KM231234	KM231412	KM231562	HQ897810	KM231705	KM232255	HQ897755	KM231959	KM232097	[18,19]
*Microcera coccophila*	CBS 310.34 = NRRL 13962	HQ897843	KM231232	KM231410	KM231560	HQ897794	KM231703	–	HQ897705	JF740692	–	[18,19,59]
*Microcera diploa*	CBS 735.79 = BBA 62173 = NRRL 13966	HQ897899	–	–	–	HQ897817	–	–	HQ897763	–	–	[18]
***Microcera kuwanaspidis***	**SICAUCC 21-0006 ^T^**	**MW462125**	**MW462126**	**MW462127**	**MW462128**	**MW484993**	**MW462905**	**MW462129**	**MW462124**	**MW462117**	**MW462130**	**This study**
***Microcera kuwanaspidis***	**SICAUCC 21-0009**	**MZ044037**	**MZ044038**	**MZ044039**	**MZ044040**	**MZ029437**	**MZ029436**	**MZ044041**	**MZ044036**	**MZ044035**	**MZ044042**	**This study**
*Microcera larvarum*	CBS 169.30	HQ897855	–	–	EU860049	EU860064	EU860064	–	HQ897717	–	EU860025	[18,60]
*Microcera larvarum*	CBS 738.79 = BBA 62239 = MUCL 19033 = NRRL 20473	KM231060	KM231230	KM231408	KM231559	KM231825	KM231701	KM232252	KM232387	KM231957	EU860026	[19,60]
*Microcera larvarum*	A.R. 4580 = CBS 133964	–	–	–	–	KC291751	KC291759	KC291894	–	KC291832	KC291935	[61]
*Microcera rubra*	CBS 638.76 ^T^ = BBA 62460 = NRRL 20475	HQ897903	KM231231	KM231409	EU860050	HQ897820	KM231702	KM232253	HQ897767	JF740696	EU860018	[18,19,60]
*Pseudocosmospora rogersonii*	CBS 133981 ^T^ = G.J.S. 90-56	–	–	–	–	KC291729	KC291780	KC291878	–	KC291852	KC291915	[61]
*Pseudocosmospora eutypellae*	CBS 133966 ^T^ = A.R. 4562	–	–	–	–	KC291721	KC291757	KC291871	–	KC291830	KC291912	[61]
*Pseudocosmospora eutypae*	C.H. 11-01 = CBS 133961 ^T^	–	–	–	–	KC291735	KC291766	KC291884	–	KC291837	KC291925	[61]
*Tilachlidium brachiatum*	CBS 505.67	KM231076	KM231249	KM231436	–	KM231839	KM231720	KM232272	KM232415	KM231976	KM232110	[19]
*Tilachlidium brachiatum*	CBS 363.97	KM231077	KM231248	KM231435	KM231583	KM231838	KM231719	KM232271	KM232414	KM231975	KM232109	[19]

Notes: superscript T represents ex-type or ex-epitype isolates. “–” means that the sequence is missing or unavailable. New sequences are listed in bold. Abbreviations. A.R.: Amy Y. Rossman, USDA-ARS, MD, USA; ATCC: American Type Culture Collection, U.S.A.; BBA: Julius Kühn-Institute, Institute for Epidemiology and Pathogen Diagnostics, Berlin and Braunschweig, Germany; CBS: Centraalbureau voor Schimmelcultures, Utrecht, Netherlands; C.H.: Cesar S. Herrera, University of Maryland, MD, USA; G.J.S.: Gary J. Samuels, USDA-ARS, MD, USA; IMI: International Mycological Institute, CABI-Bioscience, Egham, UK; MAFF: MAFF Genebank, National Institute of Agrobiological Sciences, Ibaraki, Japan; MUCL: Mycothèque de I’Université Catholique de Louvain, Belgium; NRRL: Agricultural Research Service Culture Collection, USA; TG: T. Gräfenhan collection.

## Data Availability

The datasets presented in this study can be found in the NCBI GenBank (https://www.ncbi.nlm.nih.gov/), MycoBank (http://www.MycoBank.org) and TreeBASE (http://www.treebase.org) (all accessed on 18 July 2021).

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
