# Peer review of "Insight into the Systematics of Novel Entomopathogenic Fungi Associated with Armored Scale Insect, Kuwanaspis howardi (Hemiptera: Diaspididae) in China"

_jof, 2021, doi:10.3390/jof7080628_

Round 1

Reviewer 1 Report

Dear Authors,

you can find suggestions and comments for the manuscript in the PDF attached.

The topic is very interesting, but you can improve and well organize some data. 

Author Response

Response to Reviewer 1 Comments

Point 1: My suggestion in the keywords put different words than present in the title and abstract. (Line 36)

Response 1: Yes. The keywords have been changed based on the context of the article, viz. 2 new taxa; Bamboo; Entomopathogens; Nectriaceae; Podonectriaceae; Scale insect.

Point 2: The phylogenetic tree is not very clear. My suggestion is to put a more clear figure, please. (Line 192)

Response 2: Actually, all the original pictures have high magnitude. We set the parameter of Microsoft Word, that do not compress the image in the file and keep the high resolution. And we reinsert the all the figures into the manuscript.

Point 3: What do you mean? Explain better, please (Line 239)

Response 3: The genus Tetranacrium has been recorded as the anamorph of Podonectria gahnia in 1978 by Rossman. And this section has been transferred to the discussion section of the paper.

Point 4: Can you put this information in a table? Is not clear to understand in this way, please (Line 246)

Response 4: On account of the consistent layout of the full text, the original format is preserved.

Point 5: The picture is not clear, please add another. (Line 350)

Response 5: As the answer in response 2, all the figures are re-inserted without image compression. Besides, we add other two pictures to replace the original a and c in Figure 5, that may better intelligibility.

Point 6: The discussion is very short, improve the critical point of view, please (Line 398)

Response 6: In this paper, some points have been discussed in the Notes section. But I think the point of view has been clearly discussed in this section.

Point 7: Check the references and put as request by the journal, please. (Line 484)

Response 7: Yes, the references have been checked and modified one by one, and details in the MS.

Reviewer 2 Report

Here is the review of the paper entitled "Insight into the systematics of novel entomopathogenic fungi associated with armored scale insect, Kuwanaspis howardi (Hemiptera: Diaspididae) in China" written by Xiulan Xu and co-authors.

During the research of microfungi associated with bamboo in Sichuan Province (China), the authors isolated entomopathogenic fungi associated with Kuwanaspis howardi, a scale insect on Phyllostachys heteroclada in China. Two isolated species were described as new to science (Podonectria kuwanaspidis and Microcera kuwanaspidis), while Podonectria novae-zelandiae was described for the first time from China (new record for the country). The species were described on the basis of morphological and multigene phylogenetic analyses and compared with similar species.  Phylogenetic placement of described species in Podonectriaceae (Pleosporales) is resolved.

The research methods used in the paper are appropriate and well conducted. The authors folowed all the rules needed for valid description of new fungal species. The topic is interesting for JoF audience. There are only a few suggested corrections/additions included in the attached review of the manuscript file.

After minor revision the paper deserves to be published in JoF.

Best,

Reviewer

Author Response

Point 1: There are only a few suggested corrections/additions included in the attached review of the manuscript file.

Response 1: The suggested corrections have been checked and revised.

Point 2: Briefly describe the methods here so the reader does not need to check Senanayake et al./ Briefly describe the methods here so the reader does not need to check Chomnunti et al.

Response 2: Yes, the methods have been briefly described. Details in the attached MS.
